# Effect of Concentrate Supplementation on the Expression Profile of miRNA in the Ovaries of Yak during Non-Breeding Season

**DOI:** 10.3390/ani10091640

**Published:** 2020-09-11

**Authors:** Jianpeng Xie, Qudratullah Kalwar, Ping Yan, Xian Guo

**Affiliations:** 1College of Animal Science and Technology, Gansu Agricultural University, Lanzhou 730070, China; 18709480641@163.com; 2Key Laboratory of Yak Breeding Engineering, Lanzhou Institute of Husbandry and Pharmaceutical Sciences, Chinese Academy of Agricultural Sciences, Lanzhou 730050, China; qudratullahkalwar@gmail.com

**Keywords:** miRNA, ovary, non-breeding season, yak

## Abstract

**Simple Summary:**

Yak (*Bos grunniens*) is an important and remarkable livestock species that survives in the challenging environment of the Qinghai–Tibetan Plateau. However, its growth rate is slower and reproductive ability is generally lower than cattle. This may be due to the yak living in high altitudes all year round where in the whole year, grasses are only available in July, August, and September (warm season), and from November to the next year of May (cold season), there is a scarcity of pastures. So, the reproductive efficiency of yak is very low. Meanwhile, it has been reported that enhanced nutrition improves the reproductive efficiency of animals. Therefore, this study aimed to explore the effects of supplemental nutrition on the growth traits and reproductive performance of yaks in the cold season. In addition, miRNAs related to yak reproductive traits were screened by miRNA sequencing technology. This research might be helpful for improving the reproductive potential of yak during the non-breeding season.

**Abstract:**

Yak (*Bos grunniens*) is an important and remarkable livestock species that survives in the challenging environment of the Qinghai–Tibetan Plateau. However, its growth rate is slower and reproductive potential is generally lower than cattle. Meanwhile, it has been reported that enhanced nutrition improves the reproductive efficiency of animals. The purpose of this study was to investigate the effect of concentrate supplementation on the miRNA expression profile in the ovaries of yak during the non-breeding season. The study displayed that non-breeding season supplementation significantly improved growth performance, serum biochemical indicators, and reproductive hormone concentrations in yaks. In this study, we also examined the differential expression analysis of miRNA in the ovaries of yak during non-breeding seasons using Illumina Hiseq sequencing technology. As a result, 51 differentially expressed miRNAs were found in the experimental group (CS) and control group (CON). Gene Ontology (go) and Kyoto Genome Encyclopedia (KEGG) analysis of target genes showed that beta-alanine metabolism; tryptophan metabolism; sphingolipid metabolism; alanine, aspartate and glutamate metabolism; and the inositol phosphate metabolism pathway attracted our attention. Based on qRT-PCR, seven miRNAs were assessed to verify the accuracy of the library database. We predicted and identified potential miRNA target genes, including *LEP*, *KLF7*, *VEGFA*, *GNAQ*, *GTAT6*, and *CCND2*. miRNA and corresponding target genes may regulate yaks’ seasonal reproduction through their nutritional status. This study will provide an experimental basis for improving the reproductive efficiency of yaks by supplementation in the non-breeding season.

## 1. Introduction

Yaks (*Bos grunniens*) are the only bovine animals known in the world that can survive at high elevations (average altitude of 3000 m), and they are generally scattered in the Qinghai–Tibet Plateau and its adjacent mountains or sub-alpine area [1,2]. Yaks can grow and adapt well to the alpine grassland atmosphere and breed liberally under tough plateau conditions (such as thin air, low temperature, and shortage of forage) [3]. Moreover, yaks are valuable livestock to the economy of the plateau region by providing meat, milk, and transport to the local herdsmen [4,5]. However, its growth rate is slower and reproductive ability is generally lower than cattle [6,7]. This may be due to the yak living in high altitudes all year round, where in the whole year, only grasses are available in July, August, and September, and from November to May, there is a scarcity of pastures. Therefore, the breeding rate and reproductive efficiency of yak are meager, with an average of 1 birth within 2 years, or 2 births in 3 years [8,9]. The ovary is the primary reproductive organ of female animals, and its main role is follicular development, maturation, ovulation, and synthesis and secretion of hormones to maintain normal physiological and reproductive functions [10]. Understanding the molecular regulation mechanism of the ovary, follicular development, and hormone secretion is of great significance for improving the economic benefits of livestock production.

MicroRNA is a type of endogenous noncoding small RNA of about 22 nt in length, and widely exist in eukaryotes and are highly conserved in evolution [11]. MicroRNA generally cause target mRNA degradation or translational disruption by acting on the 3 UTR of the target gene mRNA, thereby participating in the expression of gene regulation at the post-transcriptional level and playing an essential role in a series of physiological processes, such as differentiation, metabolism, cell proliferation, and apoptosis [12]. Previously, studies evaluated the miRNA expression profile of ovarian tissues of humans [13], mice [12], pigs [14], cattle [15], and sheep [16]. Several studies have shown that miRNAs are widely involved in follicular development, atresia, degradation, corpus luteum formation, and suggested that miRNAs are essential for ovarian function [17,18].

Additionally, earlier findings illustrated that supplementation during the cold season stimulates the hypothalamic-pituitary gonadal axis and enhances follicular development in prepubertal Tibetan sheep ewes on the Qinghai–Tibetan plateau [19]. Meanwhile, it has been reported that enhanced nutrition improves the reproductive efficiency of animals. Therefore, in this study, Illu-mina HiSeq sequencing technology was used to explore the influence of concentrate supplementation on the expression profiling of miRNA in the ovaries of yak during the non-breeding season. This research might help to improve the reproductive potential of yak during the non-breeding season.

## 2. Materials and Methods

### 2.1. Ethical Declaration

Before the start of the trial, all experimental animals were inspected and agreed by the institutional animal care and use committee of Lanzhou Institute of Husbandry and Pharmaceutical Sciences, China. All animals were cared for according to the guide for the care and use of laboratory animals, Lanzhou Institute of Husbandry and Pharmaceutical Sciences, China. Additionally, the present exploration was approved by the animal management and ethics committee of the Lanzhou Institute of Animal Science and Veterinary Medicine, Chinese Academy of Agricultural Sciences (Permit No. SYXK-2019-005).

### 2.2. Animals and Management

In this study, 20 non-pregnant mature yaks that had a similar age were randomly chosen and allocated into two groups (*n* = 10 each group). The duration of the study was approximately 120 days (February to May, cold season), plus an additional adaptation period (14 days) for the yaks to familiarize themselves with the staff and the diet. In addition, all animals were labeled and weighed prior to the start of the trial. Moreover, control group (CON) yaks did not receive any supplementary food, and they were kept on natural grazing conditions while, in the experimental group (CS), each yak was kept on natural grazing. In addition, they were also supplemented with 0.5 kg of concentrated supplement in the morning (08:00–09:00) and afternoon (18:00–19:00), for a total of 1 kg per day. The constituents and nutritional composition of the concentrated supplement and hay are shown in (Table 1).

### 2.3. Sample Collection

At the beginning and end of the cold season supplementary feed experiment, blood was collected from the jugular vein of 20 yaks, and serum was isolated for blood biochemical and reproductive hormone detection. Before the supplementary feed experiment, the selected animals were all about 6 years old and about the same weight. In addition, 3 yaks were selected from the CS and 3 yaks were selected from the CON and executed painlessly. Ovarian tissue samples were collected from 6 yaks for sequencing of miRNA, with 3 yaks were selected from the CS and 3 yaks from the CON. After ovarian sample collection of each animal, they were kept in liquid nitrogen for transportation and finally stored at −80 ℃.

### 2.4. Blood Biochemistry and Hormonal Concentration

The concentrations of CP (crude protein), EE (crude fat), CF (crude fiber), ash (crude ash), P (phosphorus), and Ca (calcium) in the diet were quantified by feed analysis and feed quality testing techniques. On the 0th and 120th day of the trial period, each yak was weighed in the morning prior to feeding, and the the ADG (average daily weight gain) calculated as follows:
Average Daily Gain (ADG) = (Final Weight − Initial Weight)/Days.


Albumin (ALB), total protein (TP), serum blood glucose (GLU), triglycerides (TRIG), and cholesterol (CHOL) were measured using a fully automatic bioanalysis machine. Serum follicle-stimulating hormone (FSH), gonadotropin-releasing hormone (GnRH), estrogen (E2), progesterone (PROG), and luteinizing hormone (LH) concentrations were established using ELISA kits and calibrated against standard curves using a micro plate reader.

### 2.5. Extraction of RNA and Sequencing of Small RNA

Total RNA was isolated from the ovaries of concentrate supplement group (CS) yaks and from control group (CON) yaks, by using Trizol reagent (Takara Biotechnology, Dalian, China). Afterward, low-molecular-weight RNA was isolated by using 1% agarose gel electrophoresis (PAGE) and enriched for RNA molecules in the range of 18–30 nt. The product was ligated to the 3′ and 5′ ends of the optimal temperature with a proprietary adaptor and amplified by RT qPCR. Later, sufficient product was obtained by PCR and tested by using the Agilent 2100 bioanalyzer and the ABI Step One Plus Real-time PCR System and sequencing using Illumina HiSeq (Biomarker Technologies, Beijing, China), which was tested in tissues collected from both (CS and CON) groups.

### 2.6. Analysis through Nioinformatics for Sequencing of Small RNA

The filtered sRNA sequences were mapped to the yak genome map by SOAP or bowtie software, and the expression of these sRNAs and their dissemination on the genome were analyzed and then passed through the GenBank database (http://www.ncbi.nlm.nih.gov/genbank/) and Rfam Database (http://rfam.xfam.org/), which screens and removes related sequences, such as repeating ribosomal RNA (rRNA), small interfer RNA (siRNA), small nucleolar RNA (snoRNA), nuclear small RNA (snRNA), and transport RNA (tRNA). The filtered sequence was then followed by miRBase (version 22.0) (http://www.mirbase.org/) to compare and identify identified miRNAs, and predict new miRNAs with mirdeep software (Versionv2.0.5).

### 2.7. Quantification and Differential Analysis of MicroRNA

The miRNA expression levels in each sample were counted and normalized by the TPM algorithm [20]. The TPM normalized treatment formula is as follows: TPM = Readcount * 1,000,000/over over Mappedreads. DESeq R software package was used for differential expression analysis. In the detection of differentially expressed miRNA, |log2 (FC)| ≥ 1.00 and *p*-value ≤ 0.05 were used as screening criteria.

### 2.8. Target Gene Prediction, Gene Ontology, and Pathway Analysis

Analysis of the pathways could contribute to understanding the biological functions of genes. The KEGG pathway examination was directed at the prediction results of target genes (herein after referred to as “candidate target genes”). In vivo, different genes are mutually coordinated to make their biological functions. The pathway analysis provides further insight into the biologicalfunctions of the gene. In addition, KEGG is the central public database of pathways, and the paths are categorized by significant enrichment analysis. The paths in KEGG are units, and hypergeometric tests are applied to find pathways that are significantly enriched in candidate target genes compared to the entire reference gene. We used target gene prediction for differentially expressed miRNAs, prediction with target scan and RNAhybrid software, and intersection or union as prediction. The gene ontology (GO) analysis of the potential target genes was based on the terms of the Gene Ontology database (http://www.geneontology.org/) by detecting the gene function-related biology process and putting the genes with similar functions together.

### 2.9. Validation of miRNA Expression by Quantitative Real-Time PCR

From the microRNA library, we randomly selected 7 miRNAs, and U6 was used as the internal reference gene. Afterwards, qRT-PCR was performed for expression analyses of miRNA from the ovaries of yak. The reverse primer was universal primer provided with the Mir-X™ miRNA First Strand Synthesis (Takara, Dalian, China) while the forward primers were designed based on the mature miRNA sequences (Table 2), which were obtained from miRBase (http://www.mirbase.org/), and all reactions were performed in 3 replications and the quantitative expression level of each target gene was calculated by using the threshold cycle 2^−ΔΔCt^ method [21].

### 2.10. Statistical Analysis

SPSS version 21.0 software (SPSS, Inc., Chicago, IL, USA) was used for statistical analysis. The results of body weight and blood biochemical indexes were expressed as mean ± SEM. qRT-PCR verification results were expressed as mean ± SD. An independent t-test was used for the comparison of groups. Differences were considered significant when *p* < 0.05.

## 3. Results

### 3.1. Effect of Mixed Diet Supplementation during the Cold Season on the Growth Performance of Yaks

The effects of a mixed formula diet during the cold season on the growth of yaks are presented in Table 3. At the start and end of the experiment, each yak was weighed. Weight gain in the CS was significantly higher (*p* < 0.05) than in the CON after mixed diet supplementation during the cold season.

### 3.2. Effect of Mix Diet Supplementation during the Non-Breeding Season on Serum Biochemical Parameters of the Yaks

The serum biochemical parameters of the yak were also evaluated at the start and end of the study. The concentrations of serum biochemical parameters, such as, GLU, TP, ALB, TRIG, and CHOL, were significantly (*p* < 0.01) higher in the CS than the CON (Table 4).

### 3.3. Effect of Cold Season Mix Diet Supplementation on Serum Reproductive Hormone of Yaks

The concentrations of reproductive hormones in the serum of yaks were measured at the start and after completion of the study. The results revealed that the concentrations of GnRH, FSH, LH, E2, and P in the serum of the CS were significantly higher than those in the CON (Table 5).

### 3.4. Solexa Sequencing of Ovary Small RNAs

This study was designed to understand the expression patterns of small RNA in the yak ovary during the non-breeding season following different nutrient levels, through small RNA construction library analysis. Therefore, two miRNA libraries were constructed from six yaks’ ovary samples, including concentrate supplementation (CS) yaks with an enhanced forage nutrient status and CON yaks with a lower forage nutrient status. The two libraries obtained contained 52,727,701 and 41,208,552 clean reads, respectively (Appendix A). Sequence length analysis indicated that the majority of the length was 21–23 nt (Figure 1). The Bowtie software was used to sequence clean reads with the Repbase database(http://www.girinst.org/repbase/), Silva database (http://www.arb-silva.de/), Rfam database(http://rfam.xfam.org/), and GtRNAdb database(http://lowelab.ucsc.edu/GtRNAdb/), respectively, filtering snRNA, snoRNA, rRNA, and tRNA, and a repeat sequence obtained unannotated reads containing miRNA. The sRNA annotation classification statistics are shown in Table 6 and Appendix A.

### 3.5. Identification of Differentially Expressed miRNAs in the Ovary

Sequence alignment and subsequent exploration were achieved using the specified Bos_mutus.v2.0 (http://www.ncbi.nlm.nih.gov/genome/?term=yak) as the reference genome. The Bowtie software (Version v1.0.0, Johns Hopkins University, Baltimore, MD, USA) was used to sequence the unannotated reads with the reference genome to obtain the positional information on the reference genome, which is mapped reads. For the biological characteristics of miRNA, we used the miRDeep2 software package to compare the reads on the reference genome with the known miRNA precursor sequences in the miRBase database (http://www.mirbase.org/) to identify the expression of known miRNAs. After successive filtering of these data sets, we identified a total of 1074 miRNAs (670 known miRNA and 404 novels miRNA) in the libraries (Appendix A). Amongst these miRNAs, 51 differentially expressed miRNAs were found, in which 25 miRNAs were downregulated and 26 miRNAs upregulated in the experimental group vs. the control group (Figure 2). The MA diagram provides an intuitive view of the overall distribution of the expression levels and differential multiples of the two groups (Figure 3). Systematic cluster analysis was performed to analyze the similarity of miRNAs with different expressions in the two groups of ovaries. The heat map shows the aggregation of CS and CON due to their similar expression profiles (Figure 4).

### 3.6. Validation of miRNA Expression with qRT-PCR

We performed qRT-PCR to detect the expression levels of the differentially expressed miRNAs and to confirm the reliability of the sequencing data (Figure 5). For this, we selected seven known miRNAs confirmed in the CS and CON group ovarian samples of yak. The relative expression levels of the seven selected miRNAs indicated that miR-449a, miR-205, and miR-200a were significantly upregulated in the CS than the CON group (*p* < 0.05), while miR-483, miR-200b, miR-223, and miR-2431-5p were downregulated in the CS group (*p* < 0.05). The relative expression levels of the seven differentially expressed miRNAs were consistent with the RNA sequencing results. It showed that the sequencing results were true and reliable.

### 3.7. miRNA Target Gene Prediction, GO Enrichment, and KEGG Pathway Analysis

Target gene prediction was implemented using miRNA and target scan based on newly predicted miRNAs, known miRNAs, and gene sequence evidence of corresponding species. After sequencing, 670 known miRNAs were found, but 418 known miRNAs were able to predict target genes. The remaining known miRNA did not predict target genes. These known miRNAs predicted a total of 12,485 target genes. In addition, GO enrichment analysis was used on the target genes of differentially expressed miRNAs. The differentially expressed miRNAs were enriched in 117 GO terms (*p* < 0.01), including 26 molecular functions (MFs), 20 cell component (CC), and 71 biological processes (BP) in the CS vs. the CON group (Appendix A). Meanwhile, the KEGG pathway analysis of differentially expressed miRNA target genes help us to understand the biological functions of genes. The differentially expressed miRNAs target genes had no significant enriched signal pathway in CS and CON (*p* > 0.05). Although these signaling pathways were not significant, the top 20 signal pathways contain some metabolic pathways that may be indirectly related to yak reproduction (Appendix A). These signaling pathways include beta-alanine metabolism, tryptophan metabolism; sphingolipid metabolism; alanine, aspartate, and glutamate metabolism; inositol phosphate metabolism, etc. (Figure 6). We also identified some key reproduction-related genes, such as *LEP*, *NOTCH1*, *KLF7*, *VEGFA*, *GNAQ*, *GTAT6*, and *CCND2* (Table 7).

## 4. Discussion

The blood biochemistry index can reflect the change of organism metabolism to a certain extent. The concentration of total protein and albumin in serum is closely related to protein metabolism of the body, which can reflect the protein level in the animal’s diet and the degree of digestion and absorption of protein. The results showed that the concentration of total protein and albumin in yak serum in the supplementary group was significantly higher than that in the control group. This indicated that the supplementation of concentrate increased the intake of yak protein, so the synthesis of protein in vivo increased, and the concentration of total serum protein and albumin increased. This is consistent with published research [22,23,24]. Serum triglyceride levels are closely related to fat deposition and energy metabolism in animals, which can be broken down for energy and converted to adipose tissue for storage. After the end of the experiment, the serum triglyceride content of the supplement group was significantly higher than that of the control group. The results showed that the supplementary feed increased the intake of nutrients and provided a substrate for lipid synthesis in yaks, which was beneficial to the fat deposition of the growing yaks. Blood sugar is an important component of the body and an important source of energy. The body needs a lot of sugar every day to provide energy for the normal operation of various tissues and organs to provide power. Cholesterol is an indispensable and important substance for animal tissue cells. It is not only involved in the formation of cell membranes but also the raw material for the synthesis of bile acids, vitamin D, and steroids. The results showed that the serum glucose and cholesterol levels of the supplement group were significantly higher than those of the control group.

For female reproduction, appropriate nutrition is an essential factor at all stages [25]. Under adequate nutrient conditions, follicular growth and the ovulation rate increased magnificently in livestock while these were reduced during malnutrition [26]. Meza-Herrera et al. [27] illustrated that additional supplementation in the diet improves the reproductive potential of animals. Similarly, some other findings displayed that nutritional supplementation influenced the serum biochemical parameters [28,29,30]. The current findings are also in agreement with the above findings and our results revealed that concentrate supplementation enhances the biochemical parameters and concentration of reproductive hormones.

Yak is an endemic species in high-altitude areas, and it grazes all year round. Therefore, yaks have certain wild habits. Compared with dairy cattle, sheep, and other domestic animals, yak sampling and experimental data collection is difficult. Therefore, three yaks were selected in the experimental group and the control group, respectively. Then, the miRNA sequencing analysis was carried out. Ma X et al. [31] studied the transcriptome analysis of the molecular mechanism of yak longissimus dorsi in different development stages. Three biological repeats were selected for follow-up analysis in each group. LAN D et al. [32] studied the comparative transcriptome analysis of yak and yellow cattle ovaries during estrus. Three biological replicates in each group were selected for follow-up analysis. Therefore, three biological repeats in each group were statistically significant.

Yak is an endemic species in high-altitude areas, and it grazes all year round. Therefore, yaks have certain wild habits. Compared with dairy cattle, sheep, and other domestic animals, yak sampling and experimental data collection is difficult. Therefore, three yaks were selected in the experimental group and the control group, respectively. Then, the miRNA sequencing analysis was carried out. Ma X et al. [31] studied the transcriptome analysis of the molecular mechanism of yak longissimus dorsi in different development stages. Three biological repeats were selected for follow-up analysis in each group. LAN D et al. [32] studied the comparative transcriptome analysis of yak and yellow cattle ovaries during estrus. Three biological replicates in each group were selected for follow-up analysis. Therefore, three biological repeats in each group were statistically significant.

Understanding the molecular regulation of miRNAs in the ovaries helps to explore the molecular mechanisms by which yak regulates reproduction during the non-breeding season. Many studies have shown that miRNA plays an essential role in almost all ovarian biological processes, including follicular development, follicular atresia, corpus luteum development, and degeneration [16]. This study used Solexa sequencing technology to compare the small RNA profiles of yak ovaries during the non-breeding season after concentrate supplementation. The objective was to understand the role of sRNA-mediated post-transcriptional regulation in seasonal reproduction. By obtaining miRNAs with significant differences, we found some miRNAs, such as bta-miR-483, bta-miR-449a, bta-miR-223, bta-miR-200b, bta-miR-200a, and bta-miR-2431-5p. Earlier findings illustrated that miR-483 plays a vital role in the development and regulation of livestock reproduction. Besides, another study presented that miR483 is a hormone-sensitive testis miRNA [33]. Gabriella Guelfi et al. [34] found that the reproductive efficiency of buffalo was affected by a long calving interval and late estrus. Therefore, the expression level of mir-200b in buffalo progesterone mature oocytes and pregnancy was evaluated. It was found that there were differences in the expression between pregnant and non-pregnant buffalo. Heng Yang et al. [35] explored the regulation of miRNA in sheep on seasonal reproduction and the nutritional status. It was found that mir-200b was related to the nutritional status and seasonal reproduction of sheep. Meanwhile, Sohel et al. [36] established 27 differentially expressed miRNAs, such as miR-223, between the mature oocytes and non-exosome part of follicular fluid.

In our study, RT-qPCR results showed that miR-483, miR-223, miR-200b, and miR-2431-5p were significantly downregulated in the experimental group as compared to the control group, but miR-449a, miR-205, and miR-200a were upregulated in the experimental group. These results indicate that miR-483, miR-200b, miR-2431-5p, miR-449a, miR-223, and miR-200a may be involved in the reproduction of yak through nutritional changes. In addition, the top 20 pathways, including beta-alanine metabolism; tryptophan metabolism; sphingolipid metabolism; alanine, aspartate, and glutamate metabolism; and inositol phosphate metabolism, have attracted our attention. Through KEGG pathway and software prediction, we screened the target genes of differentially expressed miRNA. Due to different nutritional levels, differentially expressed mir-483, mir-200b, mir-200a, mir-2431-5p, and mir-223 and their corresponding target genes, such as *LEP*, *VEGFA*, *GNAQ*, *GTAT6*, *CCND2*, and *KLF7*, may have potential regulatory effects on yak reproductive traits (Table 8).

Leptin may act as a metabolic signal, transmitting nutrients to the reproductive endocrine system. The hypothalamic-pituitary-gonadal axis (HPG) is activated when the body is suitable, which plays an important role in female animal estrus, fertilized egg implantation, early embryo development, pregnancy maintenance, fetal growth and development, and other reproductive activities [37,38]. Previous studies presented that leptin (*LEP*) controls the reproductive process at the periphery of the hypothalamic-pituitary and reproductive tissues (e.g., ovaries). Additionally, the leptin is expressed in the ovary, indicating that leptin and its receptors might have autocrine and paracrine effects [39]. Leptin may affect the process of steroidogenesis either indirectly or directly by modulating the actions of metabolic hormones, for example, *IGF-I GH* and insulin [40]. Meanwhile, another study displayed that leptin was associated with a shorter interval of estrus and a higher concentration of leptin was associated with it, demonstrating the relationship between leptin and estrus expression [41]. Additionally, Meikle et al. [42] displayed that in primiparous lean cows, the ovarian cycle started late, with longer intervals from birth to first conception, and the endocrine signal that most likely represents the negative energy balance of the reproductive axis is leptin. In addition, another study illustrated that leptin has different regulatory effects on the gene expression of oocytes and cumulus cells. Additionally, leptin promotes oocyte maturation and development through mechanisms independent of cumulus cells and dependent mechanisms [43]. In our study, mir-483 was upregulated in the CS group, and *LEP* was the key target gene predicted by mir-483. These results suggest that mir-483 targeting *LEP* may be involved in the reproductive process of yak.

FSH and LH are the most famous endocrine regulatory factors, which mainly promote the development and growth of follicles. However, the normal occurrence of this process also needs the participation of other factors, including vascular endothelial growth factor A (*VEGFA*) [44]. *VEGFA* is a member of the *VEGF* family of cysteine knot growth factors. Its main function is to stimulate the mitosis of endothelial cells. In the ovary, *VEGFA* was expressed in both granulosa cells and follicles as ovulation approached, and the expression of *VEGFA* in granulosa cells was significantly increased [45]. Meanwhile, some evidence shows that *VEGFA* plays an important role in follicular survival and inhibition of granulosa cell apoptosis [46]. The researchers injected *VEGFA* into the ovaries to promote the development of follicles [47]. The results of Gao et al. showed that the expression of *VEGFA* reached a peak in tertiary-generation follicles during the follicular development stage, mainly distributed in the capsule layer and the inner granular layer. During follicular development, *VEGFA* mRNA was mainly expressed in the inner layer of granules [48]. *GNAQ* is a member of G protein-coupled receptor superfamily. As an important G protein-coupled receptor, *GNAQ* promotes glucose-induced insulin release and participates in the process of nutritional metabolism [49,50]. In addition, *GNAQ* can also affect the ovary and increase the size of ovarian follicles during ovulation. *GNAQ* affects follicular growth and ovarian atresia by inhibiting PI3K-Akt signaling pathway [51,52]. Heng Yang et al. found that the effect of mir-200b differentially expressed in sheep and target gene *GNAQ* had a regulatory effect on the seasonal reproduction and nutritional status of sheep [35]. In our study, mir-200b expression was downregulated in the CS group, and *VEGFA* and *GNAQ* were key target genes predicted by mir-200b. These results suggest that mir-200b targeting *VEGFA* and *GNAQ* may be involved in yak reproductive process.

*GATA6* is a member of the *GATA* family. It plays an important role in the proliferation, differentiation, and apoptosis of ovarian cells. In the mouse ovary, *GATA6* mRNA was detected in granulosa cells and strongly expressed in the corpus luteum but not in membrane cells [53]. In the human ovary, *GATA6* mRNA was located in the granulosa cells and membrane cells of pre sinus and pre sinus follicles [54]. *GATA-6* protein was found in granulosa cells of the pig ovary [55]. Cyclins D (*CCNDs*) include three subtypes: *CCND1*/*CCND 2*, and *CCND3*. In the ovary, *CCND 2* is mainly expressed in granulosa cells. As one of the downstream factors of follicle-stimulating hormone (FSH), *CCND 2* promotes the proliferation of granulosa cells during follicular development [56]. In different stages of bovine follicular development, the expression level of *CCND 2* in granulosa cells of the active estrogen stage and preovulatory follicle was high, which was closely related to the proliferation of granulosa cells [57]. Through positive regulation of *CCND 2* expression, it can promote the cell cycle process and accelerate the proliferation of granulosa cells [58]. In our study, mir-200a was upregulated and mir-2431-5p was downregulated in the CS group. *GTAT6* and *CCND2* are key target genes predicted by mir-200a and mir-2431-5p. These results suggest that mir-200a and mir-2431-5p targeting *GTAT6* and *CCND 2* may participate in the reproductive process of yak at reproductive age.

Besides, previous studies stated that *KLF7* plays an essential role in cell proliferation, differentiation, and other processes, and participates in physiological processes, such as embryo development and glucose metabolism [59,60]. Whereas, genetic correlation analysis indicated that *KLF7* is a candidate gene for human obesity, which inhibits the differentiation of adipose precursor cells and downregulates many genes involved in adipose differentiation [61]. Overexpression of *KLF7* in already differentiated adipocytes inhibits the expression of adipocytokines, such as adiponectin [62]. In our study, *KLF7* is a key target gene predicted by mir-223. The expression of mir-223 was upregulated in the CS group. These results suggest that mir-223 targeting *KLF7* may be involved in the yak reproductive process.

## 5. Conclusions

In summary, our findings illustrated that non-breeding season supplementation significantly improved growth performance, serum biochemical indicators, and reproductive hormone concentrations in yaks. In addition, we found six differentially expressed miRNAs (mir-483, mir-449a, mir-223, mir-200b, mir-200a, mir-2431-5p). Through target gene prediction, GO enrichment, and KEGG pathway analysis, we identified that target genes corresponding to the potential miRNA may regulate the seasonal reproduction of yaks and be related to nutritional status. This study provides a new reference for the identification of new genetic markers in yaks during the non-breeding period.

## Figures and Tables

**Figure 1 animals-10-01640-f001:**
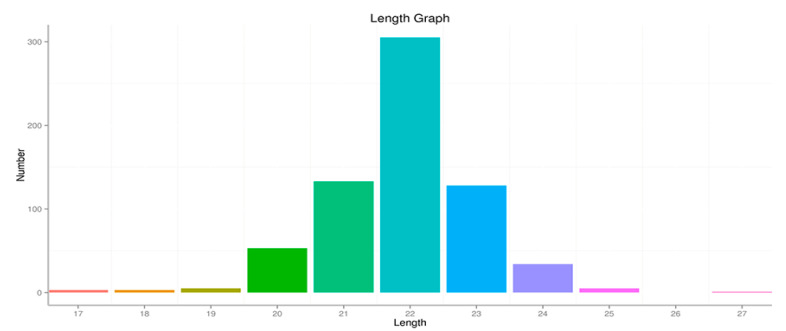
Length distribution of the sequences. (Different colors represent the distribution of miRNA length).

**Figure 2 animals-10-01640-f002:**
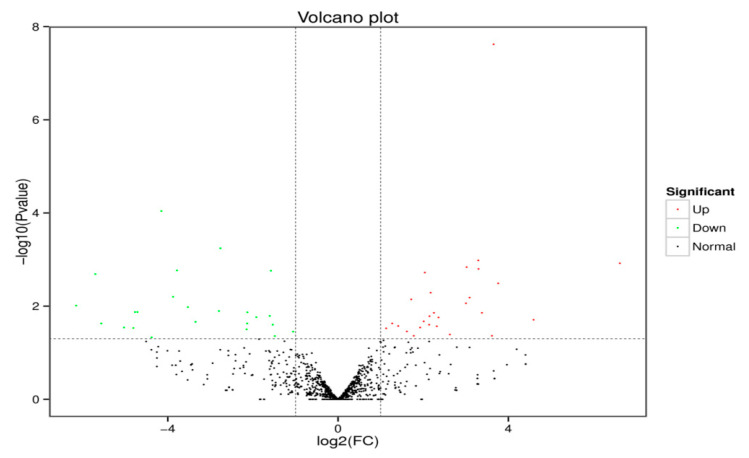
Volcano plot of the differentially expressed miRNAs in the CS and CON groups.

**Figure 3 animals-10-01640-f003:**
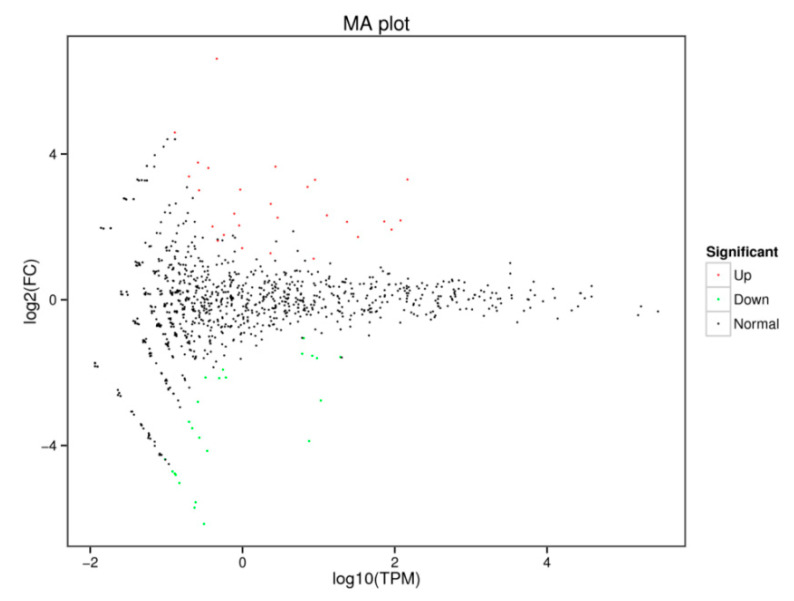
MA diagram of differentially expressed miRNA in the CS and CON groups.

**Figure 4 animals-10-01640-f004:**
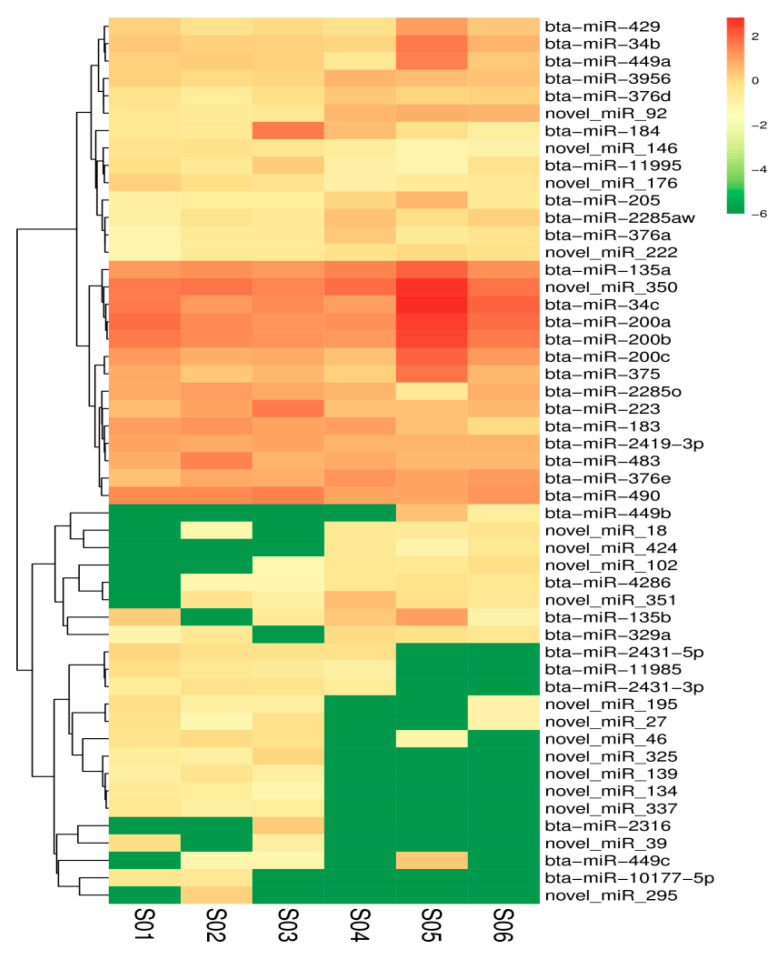
Heat map of the differentially expressed miRNAs in the CS and CON groups.

**Figure 5 animals-10-01640-f005:**
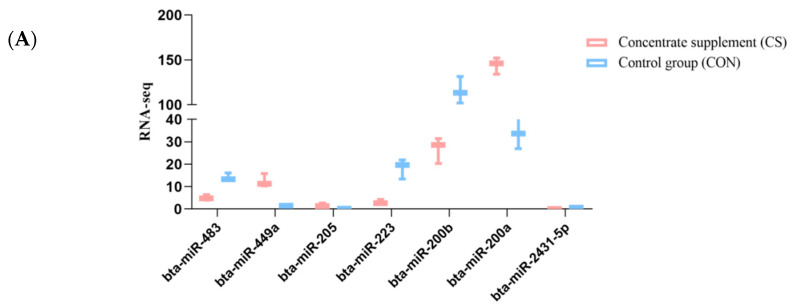
Validation for the RNA sequencing data by qRT-PCR.RNA-Seq results (**A**): qRT-PCR results (**B**): The results are presented as the mean ± SD (*p* < 0.05).

**Figure 6 animals-10-01640-f006:**
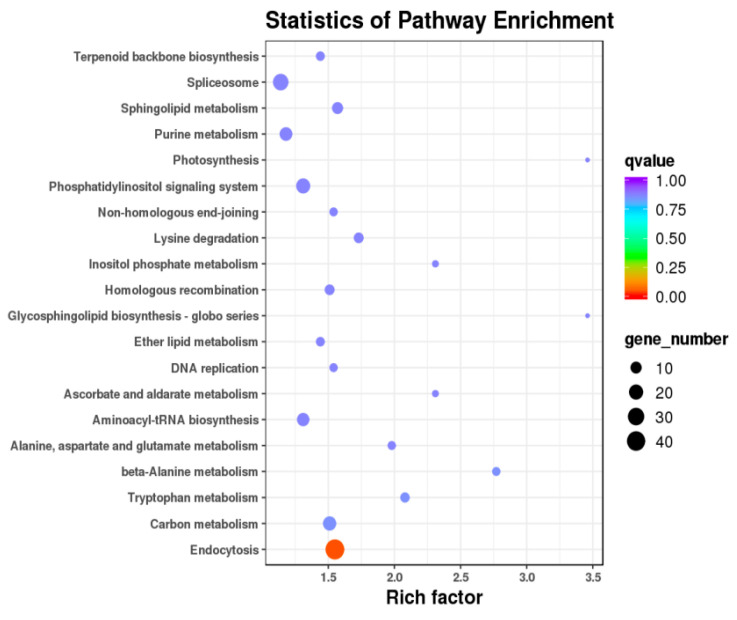
KEGG pathways of differentially expressed miRNA target genes in the CS and CON groups.

**Table 1 animals-10-01640-t001:** Nutritional composition of concentrated feed mixture and hay (%).

Constituent	Hay (%)	Concentrate Mixture (%)
Crude protein	3.65	16.74
Crude fat	1.72	3.15
Crude fiber	68.3	6.00
Crude ash	7.53	5.27
Calcium	0.54	0.84
Phosphorus	0.47	0.50

**Table 2 animals-10-01640-t002:** MiRNA primers for quantitative real-time PCR.

miRNA	miRNA Sequence (5′–3′)	Forward Primer Sequence (5′–3′)
bta-miR-483	UCACUCCUCUCCUCCCGUCUU	TCACTCCTCTCCTCCCGTCTT
bta-miR-449a	UGGCAGUGUAUUGUUAGCUGGU	TGGCAGTGTATTGTTAGCTGGT
bta-miR-200b	UAAUACUGCCUGGUAAUGAUG	TAATACTGCCTGGTAATGATG
bta-miR-200a	UAACACUGUCUGGUAACGAUGUU	TAACACTGTCTGGTAACGATGTT
bta-miR-205	UCCUUCAUUCCACCGGAGUCUG	TCCTTCATTCCACCGGAGTCT
bta-miR-223	UGUCAGUUUGUCAAAUACCCCA	TGTCAGTTTGTCAAATACCCCA
bta-miR-2431-5p	CAGGUCAUAUAAGUGUGGAGUU	CAGGTCATATAAGTGTGGAGTT

**Table 3 animals-10-01640-t003:** Effect of mixed diet supplementation during the cold season on the growth performance of the yaks.

Parameter	Groups		SEM	*p*-Value
	CON (kg)	CS (kg)		
Initial Weight	171.7 ± 24.83 ^a^	172.8 ± 31.32 ^a^	7.20	0.901
Final Weight	170.4 ± 23.88 ^a^	202.9 ± 31.75 ^b^	9.57	0.037
Average Daily Gain	−0.008 ± 0.01 ^a^	0.283 ± 0.11 ^b^	0.034	≤0.01

^a^, ^b^ values within the row showing dissimilar superscript specify a significant difference at *p* < 0.01.

**Table 4 animals-10-01640-t004:** Effect of cold season mix diet supplementation on serum biochemical parameters of the yaks.

Parameter	Days	Groups		SEM	*p*-Value
		CON	CS		
GLU (mmol/L)	0	2.84 ± 0.25 ^a^	2.87 ± 0.21 ^a^	0.06	0.761
	106	2.69 ± 0.28 ^a^	3.63 ± 0.31 ^b^	0.10	≤0.01
TP (g/L)	0	69.60 ± 4.37 ^a^	70.40 ± 3.27 ^a^	1.34	0.649
	106	66.50 ± 3.86 ^a^	84.00 ± 5.85 ^b^	1.85	≤0.01
ALB (g/L)	0	22.20 ± 1.03 ^a^	22.60 ± 1.65 ^b^	0.32	0.523
	106	21.00 ± 1.30 ^a^	32.80 ± 2.57 ^b^	0.55	≤0.01
TRIG (mmol/L)	0	0.15 ± 0.06 ^a^	0.13 ± 0.03 ^a^	0.02	0.483
	106	0.11 ± 0.03 ^a^	0.22 ± 0.06 ^b^	0.01	≤0.01
CHOL (mmol/L)	0	2.27 ± 0.37 ^a^	2.24 ± 0.31 ^a^	0.12	0.855
	106	2.11 ± 0.16 ^a^	3.59 ± 0.24 ^b^	0.08	≤0.01

^a^, ^b^ values within the row showing dissimilar superscript specify significant difference, *p* < 0.01 glucose (GLU), total protein (TP), albumin (ALB), triglyceride (TRIG), cholesterol (CHOL).

**Table 5 animals-10-01640-t005:** Effect of cold season supplementation on serum reproductive hormone of the yaks.

Parameter	Days	Groups		SEM	*p*-Value
		CON	CS		
GnRH (mIU/mL)	0	8.1875 ± 1.3287 ^a^	8.5225 ± 1.2386 ^a^	0.42	0.567
	106	7.8782 ± 1.8952 ^a^	15.9439 ± 2.8643 ^b^	0.59	≤0.01
FSH (mIU/mL)	0	1.1149 ± 0.2541 ^a^	1.1788 ± 0.1697 ^a^	0.08	0.517
	106	0.9870 ± 0.2471 ^a^	2.6079 ± 0.1806 ^b^	0.07	≤0.01
LH (mIU/mL)	0	1.0563 ± 0.2505 ^a^	1.1102 ± 0.2677 ^a^	0.07	0.648
	106	0.9431 ± 0.2780 ^a^	2.6328 ± 0.2877 ^b^	0.09	≤0.01
E2 (pg/mL)	0	10.2806 ± 4.1495 ^a^	10.9653 ± 4.2415 ^a^	0.68	0.719
	106	6.2496 ± 1.4354 ^a^	16.1687 ± 2.0088 ^b^	0.63	≤0.01
P (pmol/L)	0	450.4571 ± 50.2930 ^a^	458.0429 ± 29.6429 ^a^	9.37	0.686
	106	471.1857 ± 44.5153 ^a^	715.0143 ± 780,343 ^b^	14.0	≤0.01

^a^,^b^ values within the row showing a dissimilar superscript specify a significant difference, *p* < 0.01. Gonadotropin-releasing hormone (GnRH), follicle-stimulating hormone (FSH), luteinizing hormone (LH), estrogen (E2), progesterone (PROG).

**Table 6 animals-10-01640-t006:** Distribution of small RNA among differently expressed categories in the library

Types	CS		CON	
	Number	Percentage	Number	Percentage
rRNA	1425029	3.46%	1800091	3.41%
snRNA	130	0.00%	142	0.00%
scRNA	0	0.00%	0	0.00%
snoRNA	145152	0.35%	112544	0.21%
tRNA	89338	0.22%	72260	0.14%
Repbase	131696	0.32%	174194	0.33%
Unannotated	59020	95.65%	50568470	95.91%
Total	16362355	100%	52727701	100%

CS: The experimental group; CON: Control group.

**Table 7 animals-10-01640-t007:** The matching results of the differential miRNA and the target gene were obtained by using Target Scan software.

MiRNA-mRNA	Predicted Consequential Pairing of Target Region (Top) and miRNA (Bottom)	Site Type	Context Score	Context Score Percentile	Weighted Context Score	Conserved Branch Length	PCT
Position 1903-1909 of LEP 3′ UTR bta-miR-483	5′ ....AGUAGGGCAGAGGGCAGGAGUGU... | | | | | | | | | | | 3′ UUCUGCCCUCCUCUCCUCACU	7mer-m8	−0.16	86	−0.16	0.748	N/A
Position 190-196 of NOTCH1 3′ UTR bta-miR-449a	5′ ...UAUUUUACAUGGAAACACUGCCU... | | | | | | | 3′ UGGUCGAUUGUUAUGUGACGGU	7mer-m8	−0.49	98	−0.49	6.934	0.79
Position 2688-2694 of KLF7 3′ UTR bta-miR-223	5′ ...UUAAUUUAAUUUUUG-AACUGACC... | | | | | | | | | | | 3′ ACCCCAUAAACUGUUUGACUGU	7mer-m8	−0.21	87	−0.21	3.082	0.33
Position 1328-1334 of VEGFA 3′ UTR bta-miR-200b	5′ ...GAGUAGGGUUUUUUUCAGUAUUC... | | | | | | | 3′ GUAGUAAUGGUCCGUCAUAAU	7mer-m8	−0.23	95	−0.19	4.184	0.62
Position 4877-4883 of GNAQ 3′ UTR bta-miR-200b	5′ ...AACAUUUUUAACUUGCAGUAUUU... | | | | | | | | | | 3′ GUAGUAAUGGUCCGUCAUAAU	7mer-m8	−0.18	91	−0.15	3.766	0.53
Position 736-743 of GATA6 3′ UTR bta-miR-200a	5′ ....UUGCGUUGCAGCAAUCAGUGUUA... | | | | | | | | | | | 3′ .UUGUAGCAAUGGUCU--GUCACAAU	8mer	−0.43	99	−0.42	10.634	0.86
Position5142-5148 of CCND2 3′ UTR bta-miR-2431-5p	5′ ....ACAAUAAACUCACCUUGACCUAA... | | | | | | 3′ UUGAGGUGUGAAUAUACUGGAC	7mer-A1	−0.19	91	0.00	0	N/A

**Table 8 animals-10-01640-t008:** Reproduction-related target genes for differentially expressed miRNAs.

miRNAs	Genes
bta-miR-483	*LEP*
bta-miR-449a	*NOTCH1*
bta-miR-223	*KLF7*
bta-miR-200b	*VEGFA*
	*GNAQ*
bta-miR-200a	*GTAT6*
bta-miR-2431-5p	*CCND2*

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
