# Peer review of "Effect of Concentrate Supplementation on the Expression Profile of miRNA in the Ovaries of Yak during Non-Breeding Season"

_animals, 2020, doi:10.3390/ani10091640_

Round 1

Reviewer 1 Report

Reviewed manuscript "Effect of concentrate supplementation on the expression profile of miRNA in the ovaries of Yak during non-breeding season" (animals-913080) contains the results of very interesting research work of scientific and practical significance.
MicroRNAs (miRNAs) are a group of small molecules, about twenty nucleotides in length. They are involved in the regulation of gene expression mainly at a posttranscriptional level. This function depends on their complementarity to the 3’UTR regions of mRNAs. MicroRNAs are essential for proper development and functioning of the organism. They are so important because of their participation in such processes as angiogenesis, apoptosis, cell cycle control and oncogenesis. Over thirty percent of human genes are controlled by miRNAs. This indicates the great importance of these molecules. That's probably why the authors chose this research method
Statistical analysis of the obtained results is correct.
Tables and figures presented the results and statistical data were constructed properly.
The discussion was carried out properly and the literature used in this part of the manuscript was chosen accordingly.
However, it contains several inaccuracies:
-line 110 - "(EE) Crude fat, (CF) Crude fiber, Ash (Crude ash)- please use lowercase like "crude protein"
-line 114 - no need to explain each time what is ALB, TP, GLU etc., similarly in the case of CS and CON, please use the abbreviation and its expansion when it is first used and then one form throughout the text
-line 133 - rRNA, siRNA, snoRNA, snRNA, tRNA- when we use abbreviations their expansion should be given for the first time, like in line 206-
linne 206-207 - we only leave abbreviations
References:
-no year of publication in position 19 (line 469), position 30 (line 501), position 35 (line 515).
In summary - the manuscript contains valuable results and should be publishing in Animals

Reviewer 2 Report

The manuscript by Xie et al, entitled “Effect of concentrate supplementation on the expression profile of miRNA in the ovaries of Yak during non-breeding season" aim to provide an experimental basis for improving the reproductive efficiency of yak by supplementation in non-breeding season. The authors demonstrated that supplementation induces the differentially expression of miRNAs in experimental group. By in silico analyses the authors predicted and identified potential miRNA target genes, including LEP, KLF7, VEGFA, GNAQ, GTAT6 and CCND2 that, togheter with the corresponding miRNA may regulate yak seasonal reproduction through nutritional status. Since the study design is well structured and the results are supported by the data, the paper deserves to be accepted but some minor revisions are necessary.

1-      The main limitation of this study is represented by the low number (3) of animals used for sequencing of miRNA. This must be highlighted and discussed in the discussion section.

2-      Data presented in Figure 2 may be not clearly understood by the readers. Please use another graph type or a table to present these data.

3-      Data from qPCR must presented as whisker plots; this graphical approach is very effective and easy to read, as it can summarize data from multiple sources and display the results in a single graph

4-      Line 162. 2-△△Ct
